# TMEM120A is a coenzyme A-binding membrane protein with structural similarities to ELOVL fatty acid elongase

Jing Xue[1,2,3], Yan Han[1,2,3], Hamid Baniasadi[4], Weizhong Zeng[1,2,3], Jimin Pei[2,3,4], Nick V Grishin[2,3,4], Junmei Wang[5], Benjamin P Tu[4], Youxing Jiang[1,2,3]*

[1]Department of Physiology, University of Texas Southwestern Medical Center, Dallas, United States; [2]Department of Biophysics, University of Texas Southwestern Medical Center, Dallas, United States; [3]Howard Hughes Medical Institute, University of Texas Southwestern Medical Center, Dallas, United States; [4]Department of Biochemistry, University of Texas Southwestern Medical Center, Dallas, United States; [5]Department of pharmaceutical Sciences, School of Pharmacy, University of Pittsburgh, Pittsburgh, United States

**Abstract** TMEM120A, also named as TACAN, is a novel membrane protein highly conserved in vertebrates and was recently proposed to be a mechanosensitive channel involved in sensing mechanical pain. Here we present the single-particle cryogenic electron microscopy (cryo-EM) structure of human TMEM120A, which forms a tightly packed dimer with extensive interactions mediated by the N-terminal coiled coil domain (CCD), the C-terminal transmembrane domain (TMD), and the re-entrant loop between the two domains. The TMD of each TMEM120A subunit contains six transmembrane helices (TMs) and has no clear structural feature of a channel protein. Instead, the six TMs form an $\alpha$-barrel with a deep pocket where a coenzyme A (CoA) molecule is bound. Intriguingly, some structural features of TMEM120A resemble those of elongase for very long-chain fatty acids (ELOVL) despite the low sequence homology between them, pointing to the possibility that TMEM120A may function as an enzyme for fatty acid metabolism, rather than a mechanosensitive channel.

*For correspondence: youxing.jiang@utsouthwestern. edu

**Competing interests:** The authors declare that no competing interests exist.

## Introduction

TMEM120A was initially identified as a nuclear envelope transmembrane protein (NET) by proteomics and was originally named as NET29 (*Malik et al., 2010*; *Schirmer et al., 2003*). It was suggested to be preferentially expressed in adipose and plays an important role in adipocyte differentiation in an earlier study (*Batrakou et al., 2015*). In a recent follow-up study, the same group demonstrated that adipocyte–specific *Tmem120a* knockout mice cause disruption of fat-specific genome organization and yield a latent lipodystrophy pathology similar to lamin-linked human familial partial lipodystrophy type 2 (FPLD2) (*Czapiewski et al., 2021*). However, a completely different function has been proposed for TMEM120A in another recent study in which TMEM120A, renamed to TACAN, was shown to be expressed in the plasma membrane of a subset of sensory neurons and function as a mechanosensitive channel involved in sensing mechanical pain (*Beaulieu-Laroche et al., 2020*). This finding of a potential novel mechanosensitive channel propelled us to pursue the structural and functional studies of human TMEM120A. However, we were unable to reproduce the mechanosensitive activity of TMEM120A expressed in HEK293 or CHO cells, nor did we observe any mechanosensitive channel activity in giant liposome patching using TMEM120A protein reconstituted into lipid vesicles. Here we present the single-particle cryogenic electron microscopy (cryo-EM) structure of TMEM120A, which exhibits no obvious feature of a channel

protein. Instead, TMEM120A shares several common features with the recently determined ELOVL7 structure (*Nie et al., 2021*), a member of very long-chain fatty acid (ELOVL) family elongases important for the biosynthesis of very long-chain fatty acids. The ELOVL elongases (ELOVL1-7) are endoplasmic reticulum (ER) membrane enzymes that catalyze a condensation reaction between a long-chain acyl-coenzyme A (CoA) and malonyl-CoA to produce a 3-keto acyl-CoA, free CoA, and $CO_2$ (*Deák et al., 2019*; *Jakobsson et al., 2006*; *Leonard et al., 2004*; *Pereira et al., 2004*), which is the first step in the four-step elongation process of very long-chain fatty acids. While we are unable to define the physiological function of TMEM120A in this study, its structural similarity to ELOVL7 leads us to suspect that TMEM120A may function as an enzyme for lipid metabolism rather than an ion channel.

## Results

### Electrophysiological analysis of TMEM120A

To test if TMEM120A functions as a mechanosensitive channel, we expressed TMEM120A in HEK293 cells and measured pressure-evoked currents using patch-clamp recordings in a cell-attached configuration ('Materials and methods'). Similar pressure-evoked currents were observed in both the control cells (without transfection) and HEK293 cells expressing TMEM120A (*Figure 1a*). These pressure-evoked currents were likely from the endogenous Piezo1 channel as no pressure-elicited channel activity was observed when Piezo1 knockout (P1KO) HEK293 cells were used for TMEM120A expression and recordings (*Figure 1b*). A similar experiment was also performed using CHO cells and the same pressure-evoked background currents were observed in both the control cells and CHO cells expressing TMEM120A (*Figure 1c*). We also reconstituted the purified TMEM120A protein into lipid vesicles and employed giant liposome patching to assay the channel activity of TMEM120A under pressure ('Materials and methods'). No mechanosensitive channel activity was observed when proteoliposomes with a low protein-to-lipid ratio (1:500, w:w) were used in our patch-clamp recordings. Some transient currents were observed in patches of proteoliposomes with a higher protein-to-lipid ratio (1:100, w:w). These currents were insensitive to pressure and likely resulted from leaky liposome membranes when the protein content is high. Thus, we were unable to detect any mechanosensitive channel activity of TMEM120A in our electrophysiological assays.

### Dimeric structure of TMEM120A

Human TMEM120A was expressed in HEK293F cells using the BacMam system, solubilized in lauryl maltose neopentyl glycol (LMNG) detergent, and finally purified in digitonin detergent as a homodimer ('Materials and methods'). The single-particle cryo-EM structure of TMEM120A dimer was determined to the resolution of 3.2 Å (*Figure 2a–d*, *Figure 2—figure supplements 1–3*, and *Figure 2—source data 1*). The EM density map is of high quality, allowing for accurate model building for the major part of the protein containing residues 8–69, 80–255, and 261–335 for each subunit. In addition, electron density from a bound ligand is clearly visible within each subunit and can be modeled as a CoA molecule as will be further discussed later (*Figure 2f and g*).

Each TMEM120A subunit can be divided into two domains: the N-terminal coiled coil domain (CCD) containing CC1 and CC2 helices, and the C-terminal transmembrane domain (TMD) containing six membrane-spanning helices that form an α-helical barrel (*Figure 2c and d*). The two domains are connected by a membrane-penetrating re-entrant loop with a short helix (named re-entrant helix) on its tip. Although the cellular localization of TMEM120A, as well as its orientation in the membrane, is not clearly defined, multiple membrane protein topology prediction methods implemented in TOPCONS web server (https://topcons.net) all predicted that TMEM120A has both N- and C-termini inside (cytosolic side) (*Tsirigos et al., 2015*). We, therefore, consider the coiled coil side of the protein as the internal side and its opposite as the external side in our structural description. The transmembrane α-helical barrel enclosed a deep pocket only open to the inside but completely sealed off from the outside (*Figure 2e*). Thus, no discernible ion conduction pathway is present in the TMD of TMEM120A. A bound CoA ligand was later identified in the pocket (*Figure 2f and g*).

TMEM120A forms a tightly packed dimer with extensive dimerization interactions involving multiple parts of the protein (*Figure 3a*). Dimerization starts at CCD where the exceptionally long (~60

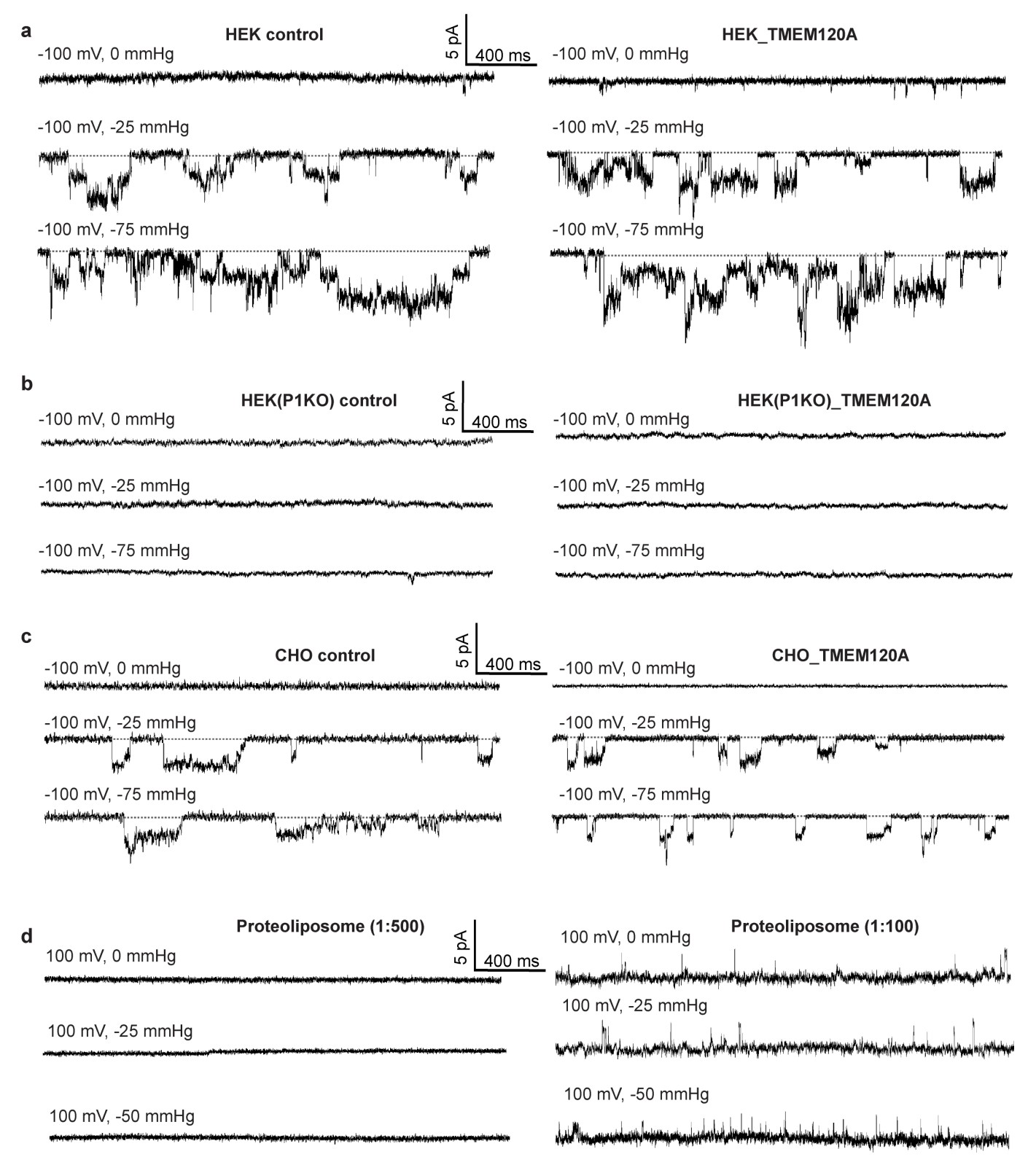

**Figure 1.** Electrophysiology of TMEM120A. (**a**) Sample traces of patch-clamp recordings of HEK293 cells with and without TMEM120A expression. (**b**) Recordings of Piezo1 knockout HEK293 cells with and without TMEM120A expression. (**c**) Recordings of CHO cells with and without TMEM120A expression. (**d**) Sample traces of giant liposome patching using proteoliposomes with 1:500 (left) or 1:100 (right) protein-to-lipid (w/w) ratio.

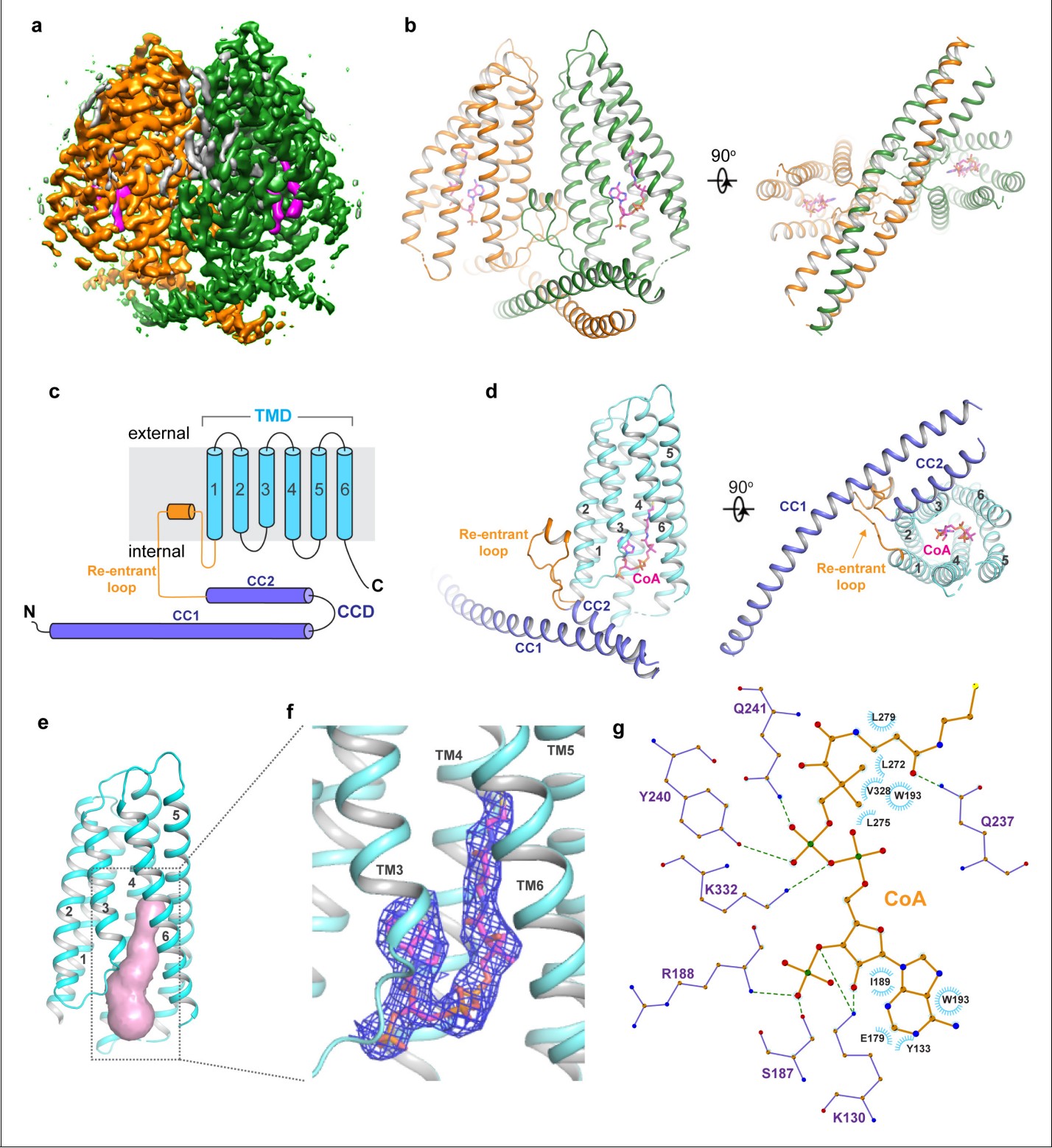

**Figure 2.** Overall structure of TMEM120A. (a) Side view of 3D reconstruction of TMEM120A. Channel subunits are colored individually with bound substrate density in purple and lipid density in gray. (b) Side and bottom views of cartoon representation of TMEM120A structure. Coenzyme A (CoA) molecules are rendered as sticks. (c) Topology and domain arrangement in a single TMEM120A subunit. (d) Side and bottom views of a single subunit in a similar orientation as the green-colored subunit in (b). (e) Transmembrane domain (TMD)-enclosed pocket (colored in salmon) analyzed using the program CAVER (*Jurcik et al., 2018*). (f) Zoomed-in view of CoA-binding site with its density (blue mesh). (g) Schematic diagram detailing the

*Figure 2 continued on next page*

*Figure 2 continued*

interactions between TMEM120A residues and CoA. Toothed wheels mark the hydrophobic contacts between protein residues and CoA. Dotted lines mark the salt bridges and hydrogen bonds.

The online version of this article includes the following source data and figure supplement(s) for figure 2:

**Source data 1.** Cryo-EM data collection and model statistics.
**Figure supplement 1.** Cryo-EM data processing scheme of human TMEM120A.
**Figure supplement 2.** Sample density maps of human TMEM120A at various regions.
**Figure supplement 3.** Sequence alignment of vertebrate TMEM120A.

residues) CC1 helix forms an anti-parallel coiled coil with CC1 from the neighboring subunit. CC2 helix has a length of about 1/3 of CC1 and runs anti-parallel to the C-terminal part of CC1, forming a 3-helix bundle with the coiled coil (*Figure 3b*). The re-entrant loop of each subunit is tightly

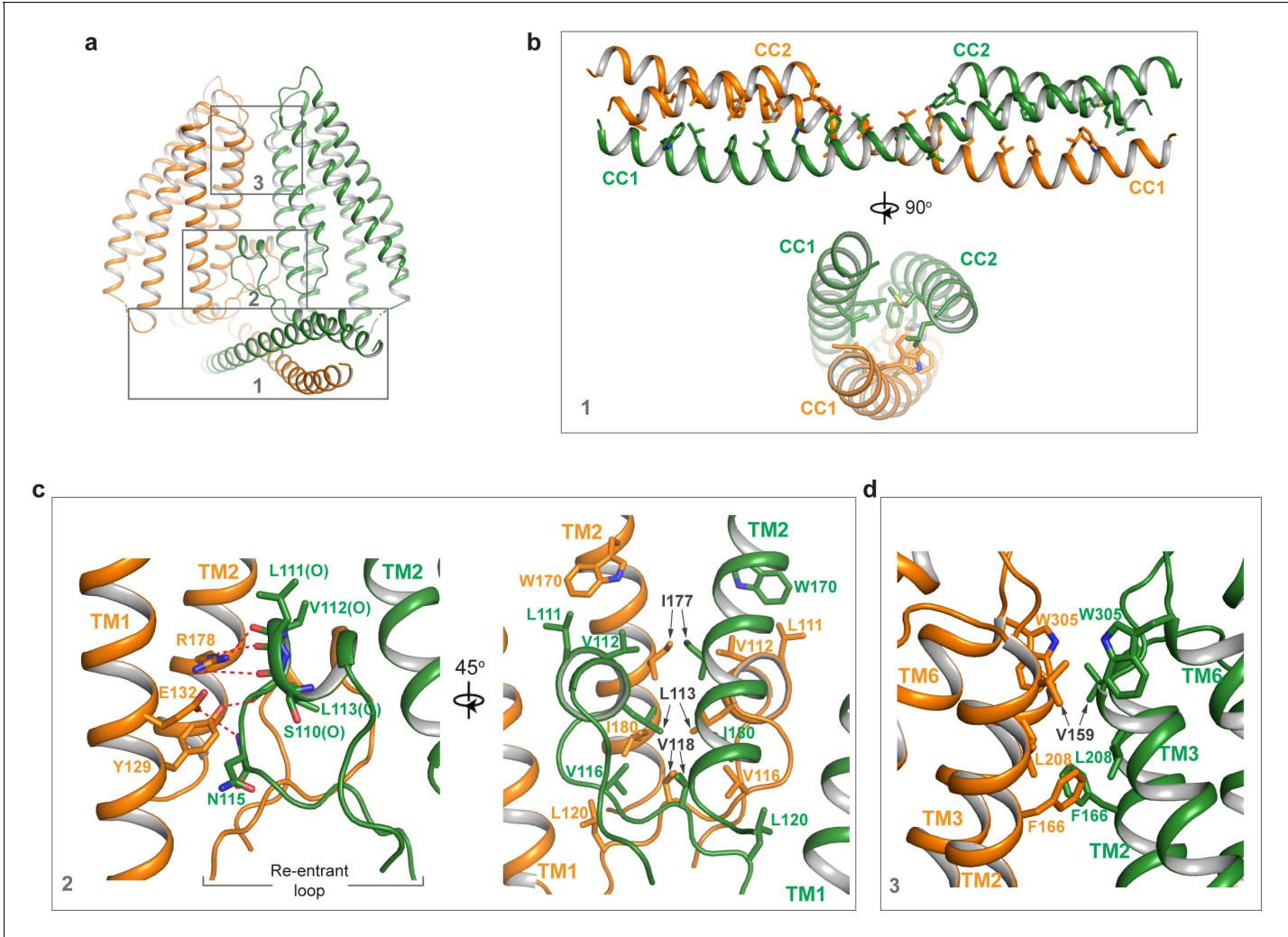

**Figure 3.** Dimerization of TMEM120A. (**a**) Extensive dimerization interactions occur in three boxed regions: the coiled coil domain (CCD) (box 1), the re-entrant loop (box 2), and the external side of transmembrane domain (TMD) (box 3). (**b**) Zoomed-in view of dimerization interactions at CCD. Residues that participate in the inter-subunit contact are W16, L19, F23, I26, H30, Y33, L37, L40, L43, I51, L58, and L61 in CC1, and L83, M87, L93, F94, M97, Y100, and L101 in CC2. (**c**) Zoomed-in view of dimerization at the re-entrant loop. Shown in the left panel are the inter-subunit hydrogen bonding interactions between R178 side chain and the carbonyl oxygen atoms of S110, L111, and V112, between Y129 side chain and the carbonyl oxygen of L113, and between the side chains of E132 and N115. Shown in the right panel are the inter-subunit hydrophobic contacts between the two re-entrant loops and between the re-entrant loop and transmembrane helices (TMs) 1-2 of the neighboring subunit. (**d**) Zoomed-in view of the inter-subunit hydrophobic contacts between the two TMDs.

wedged between the two TMDs of the dimer at the internal leaflet of the membrane. It mediates another set of extensive dimerization interactions through predominantly Van der Waals contacts with the re-entrant loop from the neighboring subunit as well as the internal halves of TMDs from both subunits (*Figure 3c*). This insertion of the re-entrant loops between the two subunits splits apart the two TMDs at the internal leaflet of the membrane and consequently the two TMDs make direct contact only at the external leaflet of the membrane through some hydrophobic residues at the external parts of TM2, TM3, and TM6, rendering the TMEM120A dimer with an arrowhead-shaped transmembrane region (*Figure 3d*). Thus, the extensive dimerization of TMEM120A involving virtually every part of the protein implies that the protein has to function as a dimer.

## Structural similarities between TMEM120A and ELOVL fatty acid elongase

The overall structure of TMEM120A shows no clear feature of a channel protein and has no discernible ion conduction pathway. We performed a structure homology search using DALI, a protein structure comparison server (http://ekhidna2.biocenter.helsinki.fi/dali/) (*Holm and Rosenström, 2010*), and identified the human ELOVL7 structure (PDB code: 6Y7F) (*Nie et al., 2021*) to share the same fold as TMEM120A at the TMD region. ELOVL7 is an ER membrane enzyme and belongs to ELOVL family elongases that catalyze the condensation reaction step in the elongation process of very long-chain fatty acids (*Jakobsson et al., 2006*). ELOVL7 contains seven TMs, and six of them (TMs 2–7) form a 6-TM $\alpha$-helical barrel that encloses a cytosol-facing pocket where a condensation reaction product of 3-keto acyl-CoA is bound (*Nie et al., 2021*; *Figure 4a*). Despite the low sequence homology, the 6-TM barrel structure of ELOVL7 is strikingly similar to that of TMEM120A (TMs 1–6) with a main-chain RMSD of about 2.5 Å between their barrel-forming 6-TM helices (*Figure 4b*). Remote homology at the TMD region between TMEM120A and ELOVL family elongases was also detected by the HHpred server for remote protein homology detection and structure prediction (*Söding et al., 2005*).

ELOVL elongases contain a highly conserved multi-histidine motif (HxxHH) important for their catalytic activity. In ELOVL7 structure, this motif is located at the beginning of TM4 with a sequence of HVFHH (*Nie et al., 2021*; *Figure 4b*). Interestingly, TMEM120A has a sequence of WVFHH at the equivalent location of the 6-TM barrel (the beginning of TM3), almost identical to the histidine motif of ELOVL7. Furthermore, ELOVL elongases bind CoA derivatives as substrates or products and in the ELOVL7 structure a bound 3-keto acyl-CoA product is identified in the deep pocket of the 6-TM barrel (*Nie et al., 2021*). In TMEM120A structure, we also observed a piece of well-resolved electron density in the pocket of the 6-TM barrel that fits well with a CoA molecule (*Figure 2f*). Indeed, this bound ligand was confirmed to be CoA by other biochemical assays as discussed in the following section.

## CoA binding in TMEM120A

To confirm the presence of CoA in the purified protein, we measured the CoA level in the protein sample using a commercially available CoA assay kit ( MAK034, Sigma-Aldrich). In this assay, CoA was used to develop a fluorescent product (Ex = 535 nm/Em = 587 nm) whose fluorometric measurement was then used to quantify CoA in the sample. As shown in *Figure 5a*, an assay of 4 mg/ml purified TMEM120A protein sample (~0.1 mM, calculated based on OD280) at various volumes yielded a CoA concentration of about 0.15 mM in the sample, matching reasonably well to the calculated concentration of CoA with 1:1 protein/ligand ratio (*Figure 5a*).

We also identified the bound CoA in TMEM120A sample using liquid chromatography-tandem mass spectrometry (LC-MS/MS). In this experiment, the bound ligand was extracted by precipitating the purified protein using methanol and the mass and fragmentation pattern of the ligand were analyzed using precursor ion (PI) scan method in MS data acquisition. Two major peaks with retention times of about 6.9 min and 7.8 min were observed during chromatographic separation (*Figure 5b*). Peak 1 was identified to be acetyl-CoA with a mass (m/z) of 810 Da when the two main product ions characteristic of acetyl-CoA fragmentation (303 Da and 428 Da) were used in the scan (*Figure 5c*). However, peak 2 exhibited a much higher intensity and was detected to have the mass of CoA (m/z = 768 Da) when using the main fragment of CoA at 428 Da in the PI scan (*Figure 5d*). A product ion scan of the 768 Da mass peak yielded the same fragmentation pattern as a CoA standard,

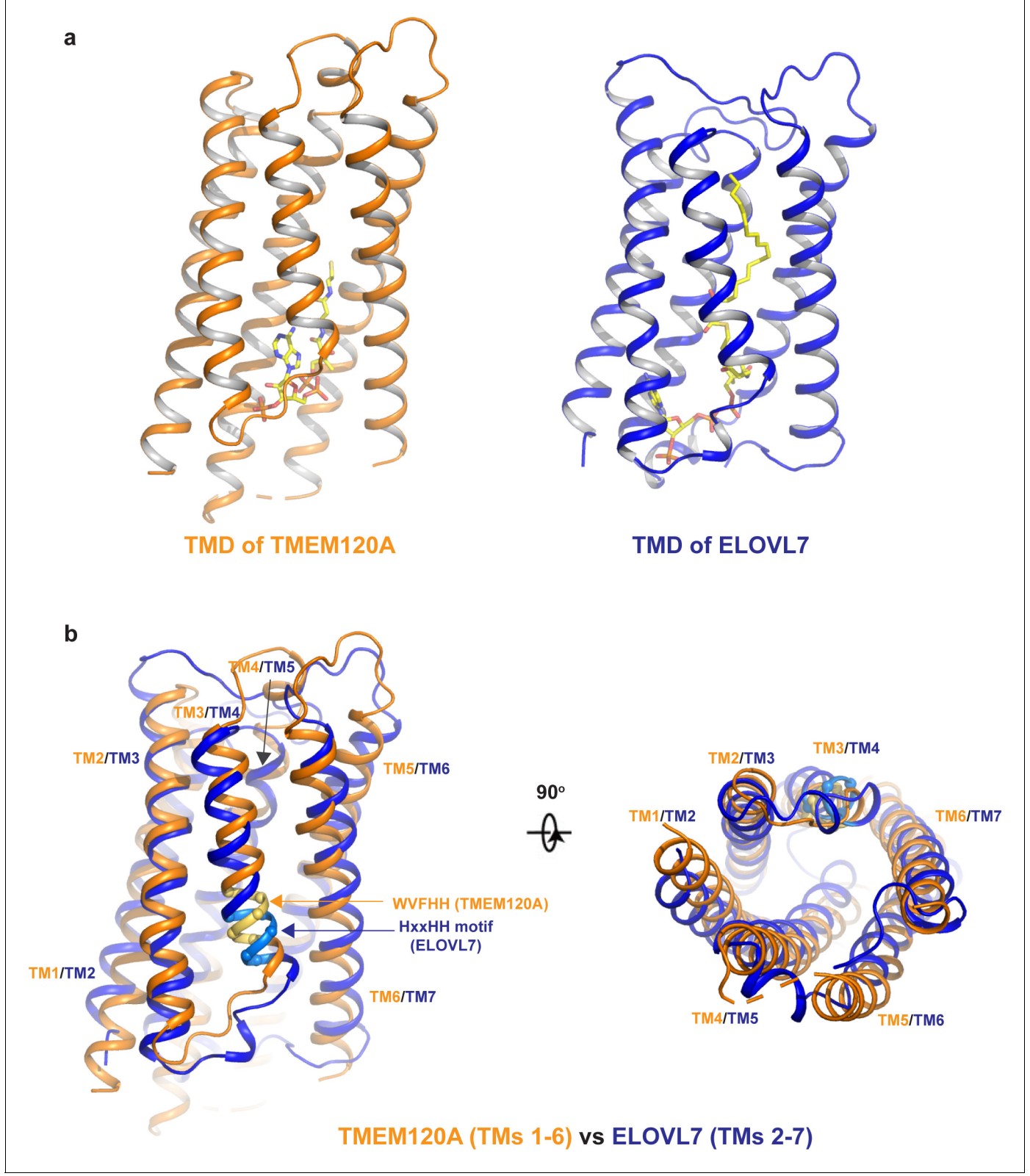

**Figure 4.** Structural comparison between TMEM120A and ELOVL7 elongase. (a) Structures of the 6-TM α-barrel transmembrane domains (TMDs) from TMEM120A (TMs 1–6, left) and ELOVL7 elongase (TMs 2–7, right). Coenzyme A (CoA) in TMEM120A and 3-keto acyl-CoA in ELOVL7 are rendered as sticks. (b) Structural comparison between the 6-TM barrels from TMEM120A (orange) and ELOVL7 (blue) in side view (left) and bottom view (right). HxxHH motif in ELOVL7 is colored in cyan. The WVFHH sequence of TMEM120A at the equivalent location is colored in yellow.

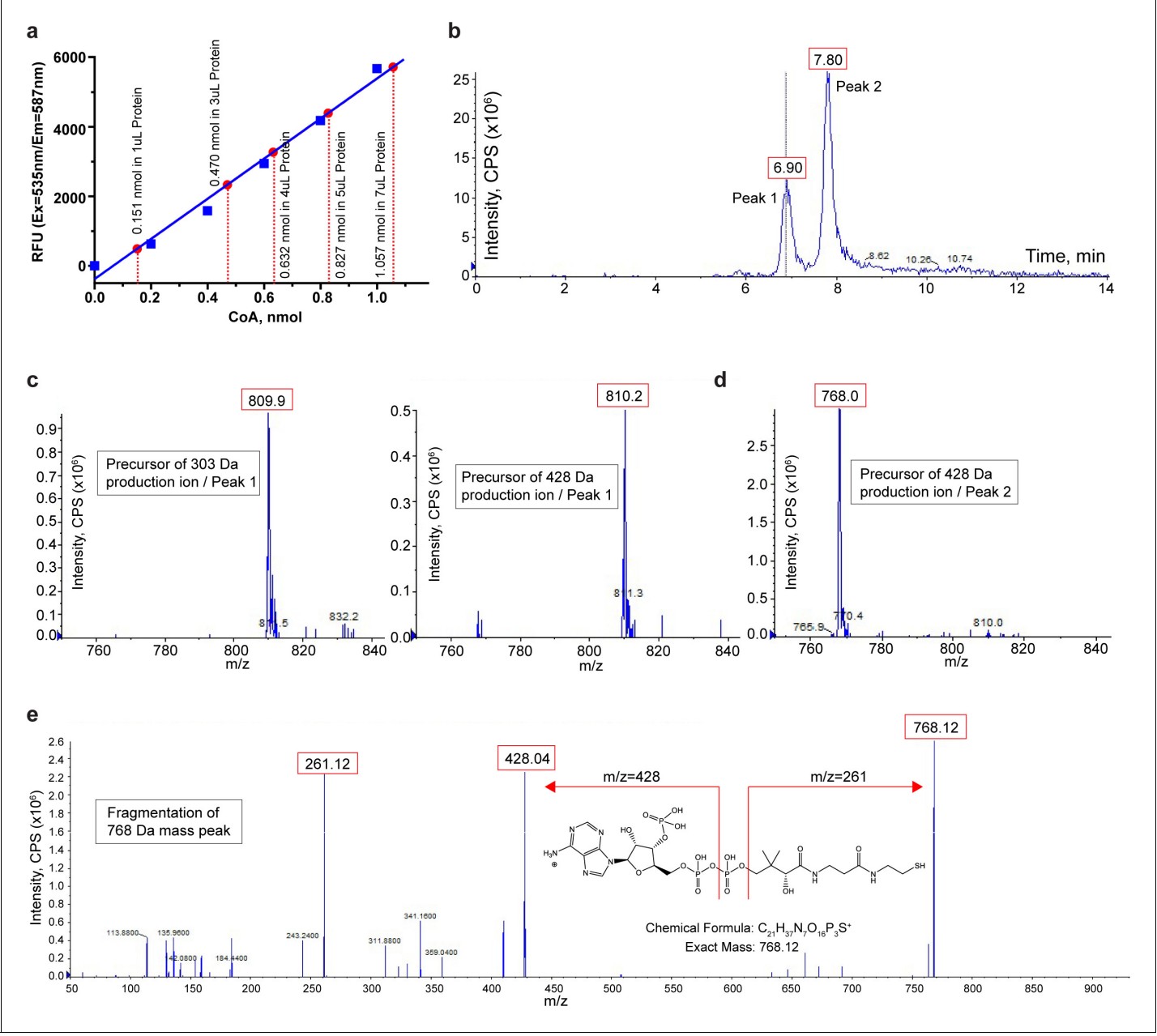

**Figure 5.** Biochemical and mass spectrometry assay of CoA in TMEM120A. (**a**) Coenzyme A (CoA) assay. A blue standard curve is obtained from fluorometric measurements of various amounts of pure CoA provided in the assay kit. Red dots mark the measured CoA contents in 1, 3, 4, 5, and 7 µl of protein samples. The measure of CoA concentration in 4 mg/ml protein sample is 0.1564 ± 0.0027 mM (mean ± SEM, n = 5). (**b**) Liquid chromatography (LC) separation of extracted substrates from TMEM120A protein sample in liquid chromatography-tandem mass spectrometry (LC-MS/MS). (**c**) Precursor ion scan of peak 1 effluent using 303 Da (left) and 428 Da (right) fragments. (**d**) Precursor ion scan of peak 2 effluent using 428 Da fragments. (**e**) Fragmentation (product ion scan) of the 768 Da mass peak.

The online version of this article includes the following source data for figure 5:

**Source data 1.** CoA assay.

confirming the identity of CoA in peak 2 (*Figure 5e*). Thus, LC-MS/MS analysis identified both CoA and acetyl-CoA in our protein sample. Combining this with our structural observation and the biochemical CoA assay, we suspect that CoA is likely the main ligand in the purified protein sample.

## Discussion

Here we present some structural and biochemical analyses of membrane protein TMEM120A, which forms a tightly packed dimer. The transmembrane domain of each TMEM120A subunit forms a 6-TM helical barrel where a CoA molecule can bind. While TMEM120A was recently proposed to function as a mechanosensitive channel, its structure shows no clear feature of an ion channel. Despite the low sequence homology, TMEM120A structure shares some striking similarities to ELOVL7, an ER membrane elongase for very long-chain fatty acids. Firstly, TMDs of both proteins contain a 6-TM α-barrel with a very similar topology and architecture. Secondly, both proteins can bind CoA or CoA derivative in the pocket of the 6-TM barrel. Thirdly, the conserved HxxHH motif important for the catalytic activity of ELOVL elongase is also present at the equivalent location in TMEM120A. Although the exact physiological function of TMEM120A remains to be determined, its similarity to ELOVL fatty acid elongase is unlikely to be coincidental and may imply enzymatic function of TEME120A for fat metabolism.

# Materials and methods

### Key resources table

| Reagent type (species) or resource | Designation | Source or reference | Identifiers | Additional information |
|---|---|---|---|---|
| Strain, strain background (*Escherichia coli*) | TOP10 | Thermo Fisher Scientific | Cat# 18258012 | |
| Strain, strain background (*E. coli*) | DH10bac | Thermo Fisher Scientific | Cat# 10361012 | |
| Cell line (*Spodoptera frugiperda*) | Sf9 cells | Thermo Fisher Scientific | Cat# 11496015; RRID:CVCL_0549 | |
| Cell line (*Homo sapiens*) | FreeStyle 293 F cells | Thermo Fisher Scientific | Cat# R79007; RRID:CVCL_D603 | |
| Transfected construct (*H. sapiens*) | pEZT-BM-TMEM120A-N$_{flag}$ | This paper | N/A | |
| Recombinant DNA reagent | pEZT-BM | DOI:10.1016/j.str.2016.03.004 | Addgene:74099 | |
| Sequence-based reagent | TMEM120A_F_primer: gatataGCTAGCCAACCGCCACCACCCGGGCCATTG | This paper | N/A | |
| Sequence-based reagent | TMEM120A_R_primer: gatataGCGGCCGCTCAATCTTTTTTTGAGCCATG | This paper | N/A | |
| Peptide, recombinant protein | Flag peptide | Sigma-Aldrich | Cat# F3290 | |
| Commercial assay or kit | Coenzyme A Assay Kit | Sigma-Aldrich | Cat# MAK034 | |
| Chemical compound, drug | Sodium Butyrate | Sigma-Aldrich | Cat# 303410 | |
| Chemical compound, drug | Lauryl Maltose Neopentyl Glycol | Anatrace | Cat# NG310 | |

*Continued on next page*

*Continued*

| Reagent type (species) or resource | Designation | Source or reference | Identifiers | Additional information |
|---|---|---|---|---|
| Chemical compound, drug | Digitonin | Acros Organics | Cat# 11024-24-1 | |
| Software, algorithm | MotionCor2 | *Zheng et al., 2017* | http://msg.ucsf.edu/em/software/motioncor2.html | |
| Software, algorithm | GCTF | *Zhang, 2016* | https://www.mrc-lmb.cam.ac.uk/kzhang/Gctf | |
| Software, algorithm | RELION | *Scheres, 2012* | http://www2.mrc-lmb.cam.ac.uk/relion | |
| Software, algorithm | Chimera | *Pettersen et al., 2004* | https://www.cgl.ucsf.edu/chimera; RRID:SCR_004097 | |
| Software, algorithm | PyMol | Schrödinger | https://pymol.org/2; RRID:SCR_000305 | |
| Software, algorithm | COOT | *Emsley et al., 2010* | https://www2.mrc-lmb.cam.ac.uk/personal/pemsley/coot; RRID:SCR_014222 | |
| Software, algorithm | MolProbity | *Chen et al., 2010* | http://molprobity.biochem.duke.edu/ | |
| Software, algorithm | PHENIX | *Adams et al., 2010* | https://www.phenix-online.org | |
| Other | Superose 6 Increase 10/300 GL | GE Healthcare | Cat# 29091596 | |
| Other | Anti-DYKDDDDK G1 Affinity Resin | GeneScript | Cat# L00432 | |
| Other | Amicon Ultra-15 Centrifugal Filter Units | Milliporesigma | Cat# UFC9100 | |
| Other | Quantifoil R 1.2/1.3 grid Au300 | Quantifoil | Cat# Q37572 | |
| Other | Cellfectin | Thermo Fisher Scientific | Cat# 10362100 | |
| Other | Sf-900 II SFM medium | Thermo Fisher Scientific | Cat# 10902088 | |
| Other | FreeStyle 293 Expression Medium | Thermo Fisher Scientific | Cat# 12338018 | |
| Other | Antibiotic Antimycotic Solution | Sigma-Aldrich | Cat# A5955 | |
| Other | Proteinase K | Thermo Fisher Scientific | Cat# EO0491 | |

## Protein expression and purification

Full-length Homo sapiens TMEM120A (HsTMEM120A, NCBI accession: NP_114131.1 ) was cloned into a modified pEZT-BM vector with an N-terminal Flag tag (*Morales-Perez et al., 2016*) and heterologously expressed in HEK293F cells using the BacMam system (Thermo Fisher Scientific). Bacmids were synthesized using *Escherichia coli* DH10bac cells (Thermo Fisher Scientific) and baculoviruses were produced in Sf9 cells using Cellfectin II reagent (Thermo Fisher Scientific). For protein expression, cultured HEK293F cells were infected with the baculoviruses at a ratio of 1:40 (virus:HEK293F, v/v) and supplemented with 10 mM sodium butyrate to boost protein expression level. Cells were cultured in suspension at 37°C for 48 hr and then harvested by centrifugation at 4000 ×*g* for 15 min. All purification procedures were carried out at 4°C unless specified otherwise. The cell pellet was

resuspended in buffer A (35 mM 4-(2-hydroxyethyl) -1-piperazineethanesulfonic acid [HEPES] pH 7.4, 300 mM NaCl) supplemented with protease inhibitors (2 µg/ml DNase, 0.5 µg/ml pepstatin, 2 µg/ml leupeptin, and 1 µg/ml aprotinin and 0.1 mM phenylmethylsulfonyl fluoride [PMSF]). After homogenization by sonication, HsTMEM120A was extracted with 1% (w/v) LMNG (Anatrace) by gentle agitation for 2 hr. After extraction, the supernatant was collected by centrifugation at 40,000 $\times g$ for 30 min and incubated with anti-Flag G1 affinity resin (Genescript) by gentle agitation for 1 hr. The resin was then collected on a disposable gravity column (Bio-Rad) and washed with 20 column volumes of buffer A supplemented with 0.05% (w/v) LMNG followed by 20 column volumes of buffer B (25 mM HEPES pH 7.4, 150 mM NaCl) supplemented with 0.06% (w/v) Digitonin (ACROS Organics). TMEM120A was eluted in buffer B with 0.06% (w/v) Digitonin and 0.2 mg/ml Flag peptide. The protein eluate was concentrated and further purified by size-exclusion chromatography on a Superdex200 10/300 GL column (GE Healthcare) in buffer B with 0.06% (w/v) Digitonin. The peak fractions were collected and concentrated to 5 mg/ml for cryo-EM analysis.

HEK293F cells (RRID:CVCL_D603) were purchased from and authenticated by Thermo Fisher Scientific. The cell lines tested negative for mycoplasma contamination.

## Cryo-EM data acquisition

Purified HsTMEM120A at 5 mg/ml was applied to a glow-discharged Quantifoil R1.2/1.3 300-mesh gold holey carbon grid (Quantifoil, Micro Tools GmbH, Germany), blotted under 100% humidity at 4°C, and plunged into liquid ethane using a Mark IV Vitrobot (FEI).

Cryo-EM data were acquired on a Titan Krios microscope (FEI) at the HHMI Janelia Cryo-EM Facility operated at 300 kV with a K3 Summit direct electron detector (Gatan), using a slit width of 20 eV on a GIF Quantum energy filter. Images were recorded with Serial EM in super resolution counting mode with a super resolution pixel size of 0.422 Å. The defocus range was set from −0.9 to −2.2 µm. Each movie was dose-fractionated to 60 frames under a dose rate of 9.2 e-/pixel/s using CDS (Correlated Double Sampling) mode of the K3 camera, with a total exposure time of 4.646 s, resulting in a total dose of 60 e-/Å2.

## Cryo-EM image processing

Movie frames were motion corrected and binned two times and dose-weighted using MotionCor2 (*Zheng et al., 2017*). The contrast transfer function (CTF) parameters of the micrographs were estimated using the GCTF program (*Zhang, 2016*). The rest of the image processing steps was carried out using RELION 3.1 (*Nakane et al., 2020*; *Scheres, 2012*; *Zivanov et al., 2018*). The map resolution was reported according to the gold-standard Fourier shell correlation (FSC) using the 0.143 criterion (*Henderson et al., 2012*). Local resolution was estimated using Relion.

Aligned micrographs were manually inspected to remove those with ice contamination and bad defocus. Particles were selected using Gautomatch (Kai Zhang, http://www.mrc-lmb.cam.ac.uk/kzhang/) and extracted using a binning factor of 3 (box size was 96 pixels after binning). 2D classification was performed in Relion 3.1. Selected particles after 2D classification were subjected to one around of 3D classification. An ab initio model was generated in Relion 3.1 and used as the reference for this 3D classification. Classes that showed similar structure features were combined and subjected to 3D auto-refinement and another round of 3D classification without performing particle alignment using a soft mask around the protein portion of the density. The best resolving classes were re-extracted with the original pixel size and further refined. Beam tilt, anisotropic magnification, and per-particle CTF estimations and Bayesian polishing were performed in Relion 3.1 to improve the resolution of the final reconstruction.

## Model building, refinement, and validation

EM map of HsTMEM120A is of high quality for de novo model building in Coot (*Emsley et al., 2010*). The model was manually adjusted in Coot and refined against the map by using the real-space refinement module with secondary structure and non-crystallographic symmetry restraints in the Phenix package (*Adams et al., 2010*).

The final structural model of each subunit contains residues 8–69, 80–255, and 261–335. Residues 1–7, 70–79, 256–260, and 335–343 were disordered in the structure. The statistics of the geometries

of the models were generated using MolProbity (*Chen et al., 2010*). All the figures were prepared in PyMol (Schrödinger, LLC) and UCSF Chimera (*Pettersen et al., 2004*).

The multiple sequence alignments were performed using the program Clustal Omega (*Sievers et al., 2011*).

## Coenzyme A quantification assay

HsTMEM120A was purified using the same protocol as described above. To release any bound CoA substrate from the protein, HsTMEM120A was subjected to protease digestion with 1 mg/ml proteinase K at 37°C for 1 hr (Thermo Scientific; EO0491). 0.2% sodium dodecyl sulfate was added to the digestion solution to stimulate the activity of proteinase K. After digestion, proteinase K was denatured by incubating the sample at 70°C for 7 min.

CoA levels in the protein solution after proteinase K digestion were quantified using a commercial CoA assay kit according to the manufacturer's protocol (Sigma-Aldrich; MAK034). CoA concentration is determined by an enzymatic assay, in which a colored product is developed and the colorimetric (OD at 570 nm) or fluorometric (Ex = 535 nm/Em = 587 nm) measurement of the product is proportional to the amount of CoA in the sample. We used fluorometric measurement in our assay for CoA quantification and its concentration was determined by comparing to a standard curve plotted using the pure CoA standard in the assay.

## Liquid chromatography-mass spectrometry analysis

HsTMEM120A sample for MS assay was purified using the similar protocol as described above with slight modification. The collected anti-Flag G1 affinity resin was washed with 20 column volumes of buffer C (25 mM HEPES pH 7.4, 180 mM NaCl) supplemented with 0.01% (w/v) LMNG. HsTMEM120A was eluted in buffer C with 0.01% (w/v) LMNG and 0.2 mg/ml Flag peptide. The protein eluate was concentrated and further purified by size-exclusion chromatography on a Superdex200 10/300 GL column (GE Healthcare) in buffer C with 0.005% (w/v) LMNG. The peak fractions were collected and concentrated to 13 mg/ml for MS analysis.

To extract the bound CoA substrate, the protein was precipitated by adding 640 ul of methanol (LC-MS grade) to 160 ul of concentrated TMEM120A sample (13 mg/ml) followed by 30 s of vortex. The sample was kept in −20°C freezer for 20 min before collecting the supernatant by centrifugation (16,400 *xg*) for 10 min at 4°C. The supernatant was filtered (0.2 micron polyvinylidene fluoride filter) before MS analysis.

LC-MS/MS analysis was conducted using a SCIEX QTRAP 6500+ mass spectrometer coupled to a Shimadzu high-performance liquid chromatography (HPLC) system (Nexera X2 LC-30AD). The ESI source was used in positive ion mode. The ion spray needle voltage was set at 5500 V. HILIC chromatography was performed using a SeQuant ZIC-pHILIC 5 µm polymer 150 x 2.1 mm PEEK-coated HPLC column (Millipore Sigma, USA). The column temperature, sample injection volume, and flow rate were set to 45°C, 5 µl, and 0.15 ml/min, respectively. HPLC solvent and gradient conditions were as follows: solvent A: 20 mM ammonium carbonate, 0.1% ammonium hydroxide; solvent B: 100% acetonitrile. Gradient conditions were 0 min: 20% A + 80% B, 20 min: 80% A + 20% B, 22 min: 20% A + 80% B, 34 min: 20% A + 80% B. Total run time: 34 min. Flow was diverted to waste for the first 5 min and after 16 min.

A PI scan in the range of 750–1250 Da was used to identify parent ions that yielded two product ions of 303 Da and 428 Da, which are characteristics of acetyl-CoA. This strategy was applied to monitor the presence of acetyl-CoA and other acyl-CoAs in the sample. An acetyl-CoA standard was used to confirm retention time and fragmentation to product ions. In addition, an EMS-IDA-EPI scan was used to fragment the mass peak observed at 768 Da, which was subsequently assigned as CoA. Data were analyzed using Analyst 1.7.1 software.

## TMEM120A reconstitution and giant liposome patching

HsTMEM120A was reconstituted into liposomes following the same protocol as previously described with some modifications (*Heginbotham et al., 1998*). Purified HsTMEM120A was mixed with azolectin solubilized in dialysis buffer (25 mM HEPES pH 7.4, 150 mM NaCl) supplemented with 0.05% DDM at protein:lipid (w:w) ratios of 1:100 and 1:500. The protein/lipid mixtures were incubated for 1.5 hr by gentle agitation and then dialyzed against 2 l of dialysis buffer at 4°C. Fresh dialysis buffer

(2 l each time) was exchanged every 18–20 hr for a total of three exchanges. Biobeads (Biorad) were added to the buffer for the final dialysis. The resulting HsTMEM120A proteoliposomes were divided into 100 µl aliquots, flash frozen in liquid nitrogen, and stored at −80°C. Single-channel currents were recorded using giant liposome patch clamp. Giant liposomes were formed by drying proteoliposomes on a clean coverslip overnight at 4°C followed by rehydration at room temperature. The standard bath solution contained (in mM) 145 KCl, 5 NaCl, 1 $MgCl_2$, and 10 HEPES-KOH, pH 7.4. The patch pipettes were pulled from borosilicate glass (Harvard Apparatus) with a resistance of 8–12 MΩ and filled with a solution containing (in mM) 145 NaCl, 5 KCl, 1 $MgCl_2$, 1 $CaCl_2$, and 10 HEPES-NaOH, pH 7.4. The giga seal (>10 GΩ) was formed by gentle suction when the patch pipette was attached to the giant liposome. To get a single layer of membrane patch, the pipette was pulled away from the giant liposome, and the patch pipette tip was exposed to air for 1–2 s. Negative pressure was applied through the patch pipette by suction measured as mmHg. Data were acquired using an amplifier (AxoPatch 200B; Molecular Devices) with the low-pass analog filter set to 1 kHz. The current signal was sampled at a rate of 20 kHz using a digitizer (Digidata 1322A; Molecular Devices) and further analyzed with pClamp 11 software (Molecular Devices).

### Cell-attached electrophysiology

1.5 µg of pEZT-BM vector containing HsTMEM120A was transfected into HEK293, CHO, or Piezo1 knockout (P1KO) HEK293 cells using Lipofectamine 2000 (Life Technology). To facilitate cell selection for patch clamp, 0.2 µg of pNGFP-EU vector containing green fluorescent protein (GFP) was co-transfected into cells (*Kawate and Gouaux, 2006*). 24–48 hr after transfection, cells were dissociated by trypsin treatment and kept in a complete serum-containing medium and re-plated on 35 mm tissue culture dishes in tissue culture incubator until recording. Patch clamp of cell-attached configuration was used to record mechanical sensitive currents. To increase the chance of observing mechanical sensitive channel currents, patch pipettes with a larger tip size (with low resistance of 1–2 MΩ when filled with the pipette solution) were used in the patch-clamp recordings. The pipette solution contained (in mM) 140 NaCl, 5 KCl, 1 $MgCl_2$, 1 $CaCl_2$, and 10 HEPES, pH 7.4. The bath solution contained (in mM) 140 KCl, 5 NaCl, 1 $MgCl_2$, 1 ethylene glycol tetraacetic acid (EGTA), and 10 mM HEPES, pH 7.4. Data were also acquired and analyzed using the same device and method as in the giant liposome patch-clamp experiment described above.

## Acknowledgements

Single-particle cryo-EM data were collected at the University of Texas Southwestern Medical Center Cryo-EM Facility that is funded by the CPRIT Core Facility Support Award RP170644 and the Howard Hughes Medical Institute Janelia Cryo-EM Facility. We thank Rui Yan at the Janelia Cryo-EM Facility for help in microscope operation and data collection. We thank Dr. A Patapoutian (HHMI/Scripps Research Institute) for providing the Piezo1 knockout HEK293 cells. This work was supported in part by the Howard Hughes Medical Institute (YJ and NVG) and by grants from the National Institute of Health (R35GM140892 to YJ, R35GM136370 to BPT, and GM127390 to NVG), the Welch Foundation (Grant I-1578 to YJ and I-1505 to NVG), and the National Science Foundation (1955260 to JW).

## Additional information

### Funding

| Funder | Grant reference number | Author |
| --- | --- | --- |
| Howard Hughes Medical Institute | | Nick V Grishin Youxing Jiang |
| National Institute of General Medical Sciences | R35GM140892 | Youxing Jiang |
| National Institute of General Medical Sciences | R35GM136370 | Benjamin P Tu |
| National Institute of General Medical Sciences | GM127390 | Nick V Grishin |

| Welch Foundation | I-1578 | Youxing Jiang |
| Welch Foundation | I-1505 | Nick V Grishin |
| National Science Foundation | 1955260 | Junmei Wang |

The funders had no role in study design, data collection and interpretation, or the decision to submit the work for publication.

## Author contributions
Jing Xue, Conceptualization, Data curation, Formal analysis, Investigation, Writing - original draft, Writing - review and editing; Yan Han, Data curation, Formal analysis, Investigation, Writing - original draft; Hamid Baniasadi, Jimin Pei, Junmei Wang, Formal analysis, Investigation; Weizhong Zeng, Formal analysis, Investigation, Writing - original draft; Nick V Grishin, Formal analysis, Supervision, Funding acquisition; Benjamin P Tu, Formal analysis, Supervision, Funding acquisition, Investigation; Youxing Jiang, Conceptualization, Formal analysis, Supervision, Funding acquisition, Investigation, Writing - original draft, Project administration, Writing - review and editing

## Author ORCIDs
Jing Xue  https://orcid.org/0000-0002-7331-1382
Benjamin P Tu  http://orcid.org/0000-0001-5545-9183
Youxing Jiang  https://orcid.org/0000-0002-1874-0504

## Decision letter and Author response
Decision letter https://doi.org/10.7554/eLife.71220.sa1
Author response https://doi.org/10.7554/eLife.71220.sa2

# Additional files

## Supplementary files
• Transparent reporting form

## Data availability
The cryo-EM density map and the atomic coordinates of the human TMEM120A have been deposited in the Electron Microscopy Data Bank under accession numbers EMD-24230 and the Protein Data Bank under accession numbers 7N7P, respectively.

The following datasets were generated:

| Author(s) | Year | Dataset title | Dataset URL | Database and Identifier |
|---|---|---|---|---|
| Xue J, Han Y, Jiang Y | 2021 | Cryo-EM structure of human TMEM120A | https://www.rcsb.org/structure/7N7P | RCSB Protein Data Bank, 7N7P |
| Xue J, Han Y, Jiang Y | 2021 | Cryo-EM structure of human TMEM120A | https://www.ebi.ac.uk/emdb/entry/EMD-24230 | Electron Microscopy Data Bank, EMD-24230 |

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
