## [Decision Letter]

**Acceptance summary:**

This paper will be of interest to the fields of ion channel and lipid metabolism. The major findings of this paper are supported by solid data and challenge the recent identification of a new membrane protein functioning as a mechanically activated ion channel.

**Decision letter after peer review:**

Thank you for submitting your article "TMEM120 is a coenzyme A-binding membrane protein with structural similarities to ELOVL fatty acid elongase" for consideration by *eLife*. Your article has been reviewed by 2 peer reviewers, and the evaluation has been overseen by Baron Chanda as the Reviewing Editor and Richard Aldrich as the Senior Editor. The reviewers have opted to remain anonymous.

Essential Revisions:

1) 'TMEM120' needs to be changed to 'TMEM120A' since the authors did not study TMEM120B in this work.

2) While the authors mention in the introduction that they fail to replicate mechanosensitive activity when expressing TMEM120A in cells or reconstituting purified TMEM120A in planar lipid bilayers, these data are not included in the manuscript. Because it is not unprecedented for ion channels to perform other functions (e.g., TMEM16F is both a scramblase and an ion channel), it is absolutely essential that these electrophysiological data are included to rule out a mechanosensory role for TMEM120A. With these data included, this paper is of high interest and appropriate for publication in *eLife*.

3) The authors claim that there is no resemblance of the TMEM120A structure to an ion channel, and no discernable ion conduction pathway. However, it would be useful for this to be explored in a more unbiased way, clearly illustrated, and thoroughly discussed; in particular, the intracellular cavity where CoA binds, perhaps using HOLE or other software, to show the size and shape of the cavity. Is this cavity large enough to permit ion entry, and could it open to the extracellular side in other conformations (for example, if CoA were not bound?).

*Reviewer #2:*

TMEM120A is a novel membrane protein that was recently proposed to function as a mechanically activated ion channel involved in pain sensation. In this study, Xue and colleagues present the single-particle cryo-electron microscopy structure of human TMEM120A. The structure shows that TMEM120A forms a dimer and each monomer contains six transmembrane helices. Interestingly, the structure resembles that of a long-chain fatty acid elongase and identifies a binding site for coenzyme A molecule. These data challenge the proposal of TMEM120A functioning as a force activated ion channel. Instead, TMEM120A seems to function as an enzyme involved in lipid metabolism. The conclusions of this paper are well supported by structural, biochemical, and mass spectrometry data.

*Reviewer #3:*

In this manuscript Xue et al., present the cyro-EM structure of the transmembrane protein TMEM120A, also recently named as TACAN. While TACAN had been proposed to be a novel mechanosensitive ion channel, here, the authors instead suggest that it bears resemblance to a fatty acid elongase, ELOVL. Strong evidence for this similarity is provided both by similar structural topology between the two proteins, as well as with biochemical experiments that demonstrate that coenzyme A is co-purified with TMEM120A. Overall, the implications of this manuscript are of highest importance, as they show evidence against a mechanosensory function for TMEM120A, as well as identify a possible alternative function that is in line with previous literature on knockout mice.

---

## [Author Response]

Essential Revisions:1) 'TMEM120' needs to be changed to 'TMEM120A' since the authors did not study TMEM120B in this work.

Corrected as suggested

2) While the authors mention in the introduction that they fail to replicate mechanosensitive activity when expressing TMEM120A in cells or reconstituting purified TMEM120A in planar lipid bilayers, these data are not included in the manuscript. Because it is not unprecedented for ion channels to perform other functions (e.g., TMEM16F is both a scramblase and an ion channel), it is absolutely essential that these electrophysiological data are included to rule out a mechanosensory role for TMEM120A. With these data included, this paper is of high interest and appropriate for publication in eLife.

As suggested, we included our electrophysiological analysis of TMEM120A in the revision. One correction in the revision is that we performed giant liposome (or GUV) patching instead of planar lipid bilayers when analyzing TMEM120A reconstituted in lipid vesicles.

3) The authors claim that there is no resemblance of the TMEM120A structure to an ion channel, and no discernable ion conduction pathway. However, it would be useful for this to be explored in a more unbiased way, clearly illustrated, and thoroughly discussed; in particular, the intracellular cavity where CoA binds, perhaps using HOLE or other software, to show the size and shape of the cavity. Is this cavity large enough to permit ion entry, and could it open to the extracellular side in other conformations (for example, if CoA were not bound?).

We analyzed the pocket using program CAVER (which gave similar result as HOLE). We are unable to identify any possible ion conduction pathway as the pocket the tightly sealed off from outside. CAVER-analyzed TMD-enclosed pocket was included in the revision.